# Comparison of the Transcriptomes and Proteomes of Serum Exosomes from Marek’s Disease Virus-Vaccinated and Protected and Lymphoma-Bearing Chickens

**DOI:** 10.3390/genes10020116

**Published:** 2019-02-05

**Authors:** Sabari Nath Neerukonda, Phaedra Tavlarides-Hontz, Fiona McCarthy, Kenneth Pendarvis, Mark S. Parcells

**Affiliations:** 1Department of Animal and Food Sciences, University of Delaware, Newark, DE 19716, USA; nnvsnath@udel.edu (S.N.N.); Phaedra@udel.edu (P.T.-H.); 2Department of Animal and Comparative Biomedical Sciences, The University of Arizona, Tucson, AZ 85721, USA; fionamcc@email.arizona.edu (F.M.); kpendarvis@pharosdx.com (K.P.)

**Keywords:** Marek’s disease, lymphoma, exosomes, vanin-1, insulin-like growth factor, acid labile subunit (IGFALS), COL22A1, cancer-associated exosomes

## Abstract

Marek’s disease virus (MDV) is the causative agent of Marek’s disease (MD), a complex pathology of chickens characterized by paralysis, immunosuppression, and T-cell lymphomagenesis. MD is controlled in poultry production via vaccines administered *in ovo* or at hatch, and these confer protection against lymphoma formation, but not superinfection by MDV field strains. Despite vaccine-induced humoral and cell-mediated immune responses, mechanisms eliciting systemic protection remain unclear. Here we report the contents of serum exosomes to assess their possible roles as indicators of systemic immunity, and alternatively, tumor formation. We examined the RNA and protein content of serum exosomes from CVI988 (Rispens)-vaccinated and protected chickens (VEX), and unvaccinated tumor-bearing chickens (TEX), via deep-sequencing and mass spectrometry, respectively. Bioinformatic analyses of microRNAs (miRNAs) and predicted miRNA targets indicated a greater abundance of tumor suppressor miRNAs in VEX compared to TEX. Conversely, oncomiRs originating from cellular (miRs 106a-363) and MDV miRNA clusters were more abundant in TEX compared to VEX. Most notably, mRNAs mapping to the entire MDV genome were identified in VEX, while mRNAs mapping to the repeats flanking the unique long (IR_L_/TR_L_) were identified in TEX. These data suggest that long-term systemic vaccine-induced immune responses may be mediated at the level of VEX which transfer viral mRNAs to antigen presenting cells systemically. Proteomic analyses of these exosomes suggested potential biomarkers for VEX and TEX. These data provide important putative insight into MDV-mediated immune suppression and vaccine responses, as well as potential serum biomarkers for MD protection and susceptibility.

## 1. Introduction

Marek’s disease virus (MDV) is a highly contagious alphaherpesvirus, belonging to the genus *Mardivirus-1*, that is the causative agent of Marek’s disease (MD). MD is a complex pathology of chickens characterized systemic immune suppression, paralysis, neurological signs and lesions, and the rapid formation of CD4^+^ T-cell lymphomas [1]. The chicken MDV-infective cycle begins with inhalation of infectious virus present in dander in the environment. In the lung, MDV infects the lung epithelium and antigen presenting cells (APCs), B cells, and macrophages [2]. Infected APCs transfer the virus to lymphoid tissues, where the virus undergoes productive-restrictive replication in its early target cells, which include B cells. B cells, and ostensibly other APCs, transfer MDV infection to CD4^+^ T cells that are the target for MDV latency and subsequent transformation.

An innate immune response mounted in response to early MDV lytic replication correlates with the establishment of latency, primarily in CD4^+^ T-cells. Transformation of latently-infected CD4^+^ T cells is due to expression of the MDV primary oncoprotein Meq, its splice-variant derivatives, and other MDV gene products (vTR, RLORF4, UL36, etc.) including viral microRNAs (miRNAs) [3,4,5,6]. T-cell lymphomas arise in various visceral and peripheral organs concomitant with profound systemic immune suppression due to the T-regulatory-like patterning of the transformed T-cells [7]. Latently-infected and transformed CD4^+^ T cells also transport the virus to the feather follicular epithelium (FFE), where the virus undergoes fully productive replication and is dispersed into the environment as desquamated epithelial cells (dander), where it can remain infectious for months, years, and perhaps decades.

In infected chickens, pathogenic field strains of MDV can cause a mortality of up to 100% in unvaccinated flocks; however, commercial losses due to MD are effectively controlled by the use of antigenically-related live attenuated vaccines. MD vaccines prevent tumor formation, but not super infection by virulent field strains [8]. While vaccine-induced, cell-mediated immunity plays a critical role in protection against MD and humoral immunity has a limited role in protection. Despite the widespread use of vaccination, how these vaccines, administered either in ovo at embryonic day 18, or via subcutaneous injection at day of hatch, pattern systemic immunity preventing tumor formation for the duration of the host’s life remains obscure.

In this regard, critical MDV immunogens against which cell-mediated immune responses are directed have not been fully defined, although earlier studies in JM16- and SB-1-infected chickens implicated cell-mediated cytotoxic response towards lytic antigens such as infected cell protein 4 (ICP4), to be crucial for genetic resistance seen in B^21^B^21^ MHC haplotype, as opposed to the susceptible B^19^B^19^ haplotype [9,10].

MDV infection in vivo occurs via cell-to-cell spread, and the mechanisms of MDV-mediated lymphomagenesis and systemic immune suppression continue to be defined. In terms of T-cell transformation, the MDV-encoded primary oncoprotein, Meq, is essential for tumor formation through transactivation of v-Jun target genes by Meq-Jun heterodimers, transrepression of lytic promoters by Meq-Meq homodimers, suppression of apoptotic pathways and the recruitment of chromatin remodeling complexes that repress lytic replication [11,12].

In addition, MDV-encoded and -dysregulated cellular miRNA expression have been associated with tumor formation. In particular, the MDV-encoded miR-155 homolog, miR-M4, targets numerous cellular genes for post-transcriptional repression (e.g., CEBP, HIVEP2, BCL2L13, JARID2, and PDCD6) and contributes to cellular transformation [13,14]. Additionally, miR-221 and -222 have been shown to target p27^Kip^ in the MDV-transformed MSB-1 cell line [15].

To further the understanding of MD vaccine-induced systemic protection, as well as MDV-induced systemic immune suppression, we performed small RNA sequencing and proteomic profiling of serum exosomes from CVI988 (Rispens)-vaccinated and protected chickens, referred to as vaccinate exosomes (VEX), and unvaccinated tumor-bearing chickens, referred to as tumor exosomes (TEX). Exosomes are a subset of extracellular vesicles that are typically 30–150 nm in size and are formed upon endosomal maturation by inward invagination and reverse budding of late endosomal limiting membrane [16]. These nanovesicles contained in the lumen of late endosome are referred to as intraluminal vesicles (ILVs), and the late endosome bearing ILVs is referred to as multi-vesicular body (MVB). Upon fusion of MVB with plasma membrane, ILVs are secreted into the extracellular environment where they are referred to as exosomes. These are secreted by virtually all cell types and their presence has been confirmed in all biological fluids including blood, urine, saliva, breast milk, aqueous humor, bronchial and peritoneal lavages, cerebrospinal and amniotic fluids. Exosomes contain biomolecules including proteins, nucleic acids, and lipids that reflect the source cell or tissue of origin. Exosomes serve as vehicles of intercellular and tissue communication by mediating short- or long-range horizontal transfer of these biomolecules. The resultant biological effect depends on the exosomal RNA and protein content whose selective incorporation varies depending upon the physiological or pathological state of the originating cells.

Investigation of the RNA and protein contents of VEX may indicate key immune signaling and oncogenic networks targeted during vaccine-induced systemic protection. Similarly, the contents of TEX may provide insight into MDV-induced immune suppression and lymphomagenesis. We hypothesize that these exosome populations encode molecular signatures for vaccine-induced protection and MD-induced immune suppression and tumor formation, respectively. Furthermore, since exosomes can function in antigen presentation during viral infections, determination of the protein content of VEX may reveal MDV target antigens against which vaccine-induced cell mediated responses are directed, as well as potential biomarkers of vaccine-induced protection [17,18,19]. Alternatively, investigation of protein content of TEX can reveal biomarkers for and mechanisms of immune suppression and tumor formation [20].

In the present study, we report the isolation of VEX and TEX from the serum of vaccinated and protected commercial layer chickens (VEX) and tumor-bearing layer and broiler chickens. These exosomes were characterized according to their size, morphology, concentration, sequence of their RNA transcriptomes (both miRNA and mRNA), the predicted mRNA targets of contained miRNAs, their proteomes, and the pathways likely to be affected by these miRNAs and proteins. This study represents the first in-depth transcriptomic and proteomic analyses of chicken serum-derived exosomes during MDV infection and provides a useful framework for successful vaccination and immunosuppression biomarker development.

## 2. Materials and Methods

### 2.1. Source of Serum Exosomes

Exosomes were purified from sera of commercial chickens used in two vaccine trials. Each vaccine efficacy trial used a natural exposure “shedder” model of infection, as described [21]. Briefly, chickens were obtained as unset commercial embryonated eggs (unvaccinated contacts, vaccinate treatments), and embryonic day 14 (E14) set embryonated eggs (shedders), from commercial hatcheries and transferred to incubators at the University of Delaware farm. All samples obtained from animals in this study were approved by the University of Delaware Institutional Animal Care and Use Committee under protocol 64R-2016-0, SOP1, approved 8/1/2016 (three-year protocol) to MSP.

Shedder chickens were eyedrop-vaccinated at hatch for Newcastle’s disease (NDV) and infectious bronchitis virus (IBV), neck-tagged for identification and inoculated intra-abdominally (IA) with 200-400 PFU of vv+MDV (strain TK-2a), and placed in a colony house (trial 1, 25 chickens per house, 8 houses), or in separate pens of a pilot broiler house (trial 2, 50 chickens per pen, 4 pens). At two weeks post-shedder placement, unvaccinated and MD-vaccinated chickens were eyedrop vaccinated for NDV/IBV, neck-tagged for identification, and placed in contact with the shedder chickens.

The chickens used in trial 1 were HyLine W36 white leghorns (obtained from the HyLine hatchery, Elizabethtown, PA, USA). Trial 1 consisted of a comparison of three commercial CVI988 (Rispens) vaccines with an experimental, attenuated MDV-1 vaccine. Chickens were vaccinated subcutaneously at hatch with 1× commercial dose (target dose 3500 PFU) of one of the four vaccines. Vaccinated chicks were co-housed with shedder and unvaccinated chickens, provided with commercial feed and water *ad libitum*, and spray-vaccinated against NDV/IBV on the 29th day post-placement. Birds were monitored daily for MD clinical signs and moribund chickens were culled. For each house, the target numbers of shedders, unvaccinated- and vaccinated-contact-exposed chickens were 25, 15, and 65, respectively. Surviving shedder chickens were removed at 42 days post-placement (four weeks post-mixing with other groups), euthanized, and scored for MD lesions. The trial was terminated at 49 days post-placement of vaccinates and remaining unvaccinated and vaccinated-chickens were euthanized and scored for MD lesions.

The chickens used in trial 2 were commercial broilers obtained from Mountaire Farms, Inc. (Millsboro hatchery, Millsboro, DE, USA). Trial 2 consisted of a comparison of herpesvirus of turkeys (HVT) commercial and experimental vaccines, administered in ovo at E18, at two dosage levels. This study was conducted in a pilot-scale broiler house at the University of Delaware farm (maximum capacity = 2500 broiler chickens), with the single-room being split into [4] pens separated by a central staging area. The target numbers for each pen were: shedders (20), unvaccinated contacts (15), vaccine treatments (total of 8 treatments, 25 per treatment) for a total of 285 chickens per pen (1140 for the entire study). Timing for vaccine efficacy trial 2 was identical to that of trial 1 (shedders euthanized at 42 dpi, unvaccinated and vaccinated groups terminated at 49 days post-placement). Vaccine efficacy trials were approved by the University of Delaware IACUC and were performed under protocol #64R-2016-0.

For the purposes of exosome isolation, chickens showing clear signs of MD (red-leg, ataxia, paralysis) were bled via cardiac puncture, immediately euthanized and scored for MD lymphomas (trials 1 and 2). Similarly, vaccinated and visibly-protected chickens (large, lesion-free, good body condition and coloration) were similarly bled, euthanized and scored for MD lesions (trial 1 only).

Blood samples (~10 mL) from tumor-bearing chickens were used for exosome isolation and designated as tumor-associated exosomes (TEX). Similarly, blood samples from vaccinated and visibly-protected chickens were used for exosome isolation and designated as vaccine-associated exosomes (VEX). For both groups, whole blood was collected without anticoagulant using 10 cc syringes with 18-gauge needles, and syringes were stored at 37 °C for 1 h followed by 4 °C, overnight. Serum samples were carefully collected and stored at −80 °C until exosome purification.

For in-depth characterization, TEX were isolated from MD gross lesions-positive (MD+), leghorns (n = 4) and broilers (n = 4) from efficacy trials 1 and 2, respectively. VEX were isolated from protected leghorns (n = 4) that were scored negative for MD gross lesions (MD-), from trial 1 only. Bird neck tag numbers of the chickens used for these studies are given in Table 1.

### 2.2. Serum Exosome Purification

Exosome purification was carried out using total exosome isolation reagent (TEI, Invitrogen, Carlsbad, CA, USA) solution according to the manufacturer’s recommendations as described previously [22]. For small RNA sequencing experiments, serum exosomes were purified by ExoQuick Solution (System Biosciences, Palo Alto, CA, USA), following manufacturer’s recommendations.

### 2.3. Nanoparticle Tracking Analysis

The concentration and size distribution profiles of exosome particles were evaluated by tracking their Brownian motion using a Nanosight nanoparticle tracking instrument (NS300, Malvern, Worcestershire, UK) and analyzed with NTA 3.2 Dev Build 3.2.16 software, essentially as described [22]. The following post-acquisition analysis settings were selected: minimum detection threshold 4, automatic blur, and automatic minimum expected particle size. Each sample was diluted 1:20 in particle-free PBS before injecting into the instrument. During the instrument run, three 1 min videos were recorded at camera level 12 with minimum expected particle size, track length, and blur setting, all set to default and analyzed in batch processing mode. Analyses of representative NTA histograms for VEX and TEX are shown in Figure 1C.

### 2.4. Transmission Electron Microscopy

For TEM analyses, exosome-PBS suspensions from two randomly selected samples, one VEX (Bird tag# BL3825; leghorn VEX) and one TEX (Bird tag# OR1760; leghorn TEX) were floated on to 400 mesh, carbon-coated copper grids that were previously glow discharged with a PELCO easiGlow™ glow discharge system to render the surface of the grids hydrophilic. Grids were then subjected to washes on drops of water, and then negative stained with 2% uranyl acetate. Air-dried grids were examined with a Zeiss Libra 120 transmission electron microscope at 120 kV, and images were acquired with a Gatan Ultrascan 1000 2k × 2k CCD camera in the bioimaging core at the Delaware Biotechnology Institute at the University of Delaware. For each set of analyses, at least 20 fields were imaged and representative images are shown in results sections.

### 2.5. Exosomal Small RNA Sequencing

For high throughput sequencing, LC-Sciences, LLC (Houston, TX, USA) was contracted for exosome isolation, RNA purification, library generation, sequencing, and initial bioinformatic analysis. For sequencing of exosome small RNAs, sera from vaccine efficacy trials 1 (leghorn VEX [n = 4] and leghorn TEX [n = 4]) and 2 (broiler TEX [n = 4]) were subject to precipitation by ExoQuick Solution (System Biosciences), followed by loading of exosome lysate onto SeraMiR columns (System Biosciences). Eluted RNAs were resuspended in RNAse free water. The quantity and quality of the RNA were determined by Agilent 2100 Bioanalyzer with a total RNA Pico Chip (Agilent Technologies, Santa Clara, CA, USA).

#### 2.5.1. Small RNA Library Preparation

Small RNA libraries were prepared by Illumina Truseq™ Small RNA preparation kit according to manufacturer’s instructions (Illumina, Inc., San Diego, CA, USA). Briefly, small RNAs were adaptor ligated on both 5′ and 3′ ends, followed by reverse transcription with a 3′ adaptor primer. The resultant complementary DNA (cDNA) products were subject to PCR enrichment using a common PCR Primer complementary to 5′ end adaptor and a reverse primer containing a primer index on the 5′ end of the region complementary to 3′ adaptor. The PCR enrichment step allowed library amplification along with index bar coding to facilitate multiplexing during sequencing. The indexed libraries were PCR purified and post PCR purified library was quantified on a Qubit 2.0 fluorometer (Invitrogen).

The purified cDNA library was subject to cluster generation on Illumina cluster station and then sequenced on Illumina Hiseq Platform. Raw sequencing reads (~50 nt) were obtained using Illumina’s Sequencing Control Studio software version 2.8 (SCS v2.8) following real-time sequencing image analysis and base-calling by Illumina’s Real-Time Analysis version 1.8.70 (RTA v1.8.70). A series of digital filters were applied to exclude unmappable low complexity reads (reads lacking a 3′ adaptor, adaptors with greater than one mismatch, reads >32 nts after 3′ adaptor removal, reads <15 nts after 3′ adaptor removal) from the raw sequence data to obtain unique reads. The extracted sequencing reads were stored in FastQ (*fq)* files, which were then used in the post processing steps by standard data analyses pipelines, described below.

#### 2.5.2. Small RNA seq Data Analyses

Briefly, quality control of raw reads was performed using FastQC (Babraham Bioinformatics, London, UK), and unique reads were imported into a proprietary data analysis platform, ACGT101-miR v4.2 (LC Sciences, Houston, TX, USA) for data analysis. Post-sequencing removal of adaptor sequences, low-quality reads, and common RNA families (rRNA, tRNA, snRNA, snoRNA), unique sequences of 15–32 bases were mapped to *Gallus gallus* precursor and mature miRNAs in miRBase 21.0 using a BLAST search to identify known miRNAs. For the unmapped sequences, a BLAST search was performed against the *Gallus gallus* reference genome, and the mapped sequences that contained potential hairpin RNA structures were predicted from the flanking 80-nt sequences using RNAfold software [23,24].

#### 2.5.3. MiRDeep2 Analyses of Precursor and Mature MDV-1 miRNAs

For determining precursor and mature miRNA reads originating from MDV-1 genome, a UNIX shell installed with PERL-based package, miRDeep2 (version 2.0.0.7), created by Friedlander et al. [25]*,* was utilized. Briefly, unique reads in the FASTA (*.fa*) format were filtered to limit the read length to 17 bases. Next, using miRDeep2 mapper function (*mapper.pl*), reads comprising identical sequence were collapsed to remove redundancy. Collapsed reads were mapped to the pRB1B reference genome (GenBank Accession no: EF523390) with bowtie options; bowtie –f –n 0 –e 80 –l 18 –a –m 5 –best –strata. These options correspond to non-allowance of mismatches in the seed region, allow 2 mismatches after the seed region to account for non-templated additions, and preservation of reads that map no more than 5 times in the genome. In addition, if mappings with 0 mismatches occurred, then mappings with one or two mismatches were not accounted. Finally, processed reads and the mappings to the genome were outputted in *.arf* format and directly inputted to the miRDeep2 module to identify MDV-1 miRNA read numbers.

#### 2.5.4. miRDB Prediction of miRNA Gene Targets

The set of genes targeted by each exosomal miRNA was predicted using the miRDB online resource and analysis platform (http://www.miRDB.org//). Launched in 2008, it was comprehensively updated where the complete set of miRNA sequences from the miRBase repository was downloaded along with the complete set of *G. gallus* 3′UTR sequences contained in the NCBI RefSeq database [26]. In addition, the miRDB target prediction algorithm, MiRTarget, which was developed using support vector analysis of high throughput expression data, predicts conserved and non-conserved target genes via weighting target site conservation as a high priority, but not as a strict requirement. miRDB target scores range between 50 to 100, with a greater score indicating a greater statistical confidence in the target prediction. Predicted targets with a score >80 were considered to be the most confident gene predictions and were therefore used for gene ontology and pathway enrichment analysis.

#### 2.5.5. Gene Ontology and Pathway Enrichment Analysis

The DAVID (Database for Annotation, Visualization and Integrated Discovery) [27,28] database was used to perform gene ontology enrichment analyses on miRDB predicted gene targets of VEX- or TEX-upregulated miRNAs [29]. *G. gallus* genes were uploaded into the DAVID database and enriched gene ontology terms and KEGG pathways were identified.

#### 2.5.6. Geneious Mapping of Reads to the MDV Genome 

Quality controlled unique reads were mapped against pRB1B reference genome (Accession no: EF523390) to produce a contig using the Geneious (v.10) read mapper with 10% allowed gaps per read, word length of 18, and 20% maximum mismatches per read and with structural variant, insertion, and gap finding allowed.

#### 2.5.7. Read Count Normalization and Comparisons

Normalization of read counts in each sample (or data set) was achieved by dividing the read counts by a library size parameter of the corresponding sample. Reads were removed if the corresponding maximum number of raw reads in all samples was less than 10. In a statistical test, a read was removed if the mean value of the normalized reads of all samples involved in the test was less than 10. The differential expression of each miRNA was calculated based on the fold change observed between different group comparisons (leghorn VEX versus TEX, and leghorn VEX versus broiler TEX). The “calculate expression levels” feature of Geneious was used to measure normalized expression levels of MDV-coding regions. This feature normalizes mapped raw read count by transcript length and sequencing depth and expressed as “reads per kilobase transcript per million mapped reads (RPKM)”. Statistical significance between the two groups was determined by unpaired Student’s *t*-test.

### 2.6. Proteomic Analysis of Exosomes

#### 2.6.1. Sample Preparation

For proteomic analysis, 50 μg of exosome protein from TEI-purified exosomes was submitted to Dr. Fiona McCarthy and Mr. Ken Pendarvis at the University of Arizona Core Proteomics facility (Keating Bioresearch Building, U. Arizona, Tucson, AZ, USA). Exosome protein purification was performed as described by McCarthy et al. [30]. Differential detergent fractionation was performed with digitonin followed by Triton X-100 to extract soluble and membrane associated exosomal proteins. Protein fractions were treated with a mixture of DNase I (50U; Invitrogen) and RNase A (50 mg; Sigma-Aldrich, St Louis, MO, USA) to digest intact nucleic acids at 37 °C for 30 min. Samples were centrifuged at 6200× *g* for 10 min at 10 °C to pellet the debris.

Exosomal proteins were precipitated with an equal volume of 50% trichloroacetic acid (TCA) at −20 °C overnight. The precipitated proteins were pelleted by centrifugation, washed with ice-cold acetone, and dried at room temperature. Proteins were resuspended in 0.5 mL of solubilization solution (7 M urea, 20 mM Tris-Cl, pH 8.0, 5 mM EDTA, 5 mM MgCl_2_, 4% CHAPS, 1 mM PMSF) and quantitated using the 2-D Quant Kit (GE Healthcare Life Sciences, Pittsburgh, PA, USA). An aliquot of 0.1 mg of protein was precipitated again with 50% TCA at −20 °C, followed by centrifugation and a wash with ice-cold acetone. Samples were resuspended in 0.1 mL of 100 mM ammonium bicarbonate and 5% acetonitrile, reduced with 5 mM dithiothreitol for 10 min at 65 °C, alkylated with 10 mM iodoacetamide for 30 min at 30 °C, and finally digested with 2 μg of sequencing grade trypsin at 37 °C for 16 h. Peptides were desalted using a peptide macrotrap (Michrom Bioresources, Inc., Auburn, AL, USA), dried at room temperature, and stored at −80 °C until further processing.

Desalted peptides were vacuum centrifuged until dry and resuspended in 2 μL of 2% acetonitrile, 0.1% formic acid and transferred to low retention vials in preparation for analysis using one dimensional LC–MS/MS. Samples were transferred to low retention HPLC vials for analysis using mass spectrometry. Peptide mass spectrometry was performed upon separation of proteins using a one-dimensional Dionex U3000 splitless nanoflow HPLC system operated at 333 nL per minute using a gradient from 2 to 50% acetonitrile over 4 h. The C18 column, an in-house prepared 75 μm × 10 cm reverse phase column packed with Halo 2.7 μm, 90 A° C18 material (MAC-MOD Analytical), was located in the ion source just before a Proxeon ES562 40 mm, 30 μm id stainless steel emitter. U3000 eluate was analyzed with an LTQ Velos (Thermo Scientific, Waltham, MA, USA) linear ion trap mass spectrometer.

Scan parameters for the LTQ Velos Pro were one MS scan followed by 10 MS/MS scans of the 5 most intense peaks. MS/MS scans were performed in pairs, a CID fragmentation scan followed a HCD fragmentation scan. All scans were performed in enhanced resolution mode. Dynamic exclusion was enabled with a mass exclusion time of 3 min and a repeat count of 1 within 30 s of initial m/z measurement.

#### 2.6.2. Protein Identification

Mass spectra and tandem mass spectra were used to search subsets of the non-redundant protein database (nrpd) downloaded from the National Center for Biotechnology Institute (NCBI; 06/14/16) using TurboSEQUEST (Bioworks Browser 3.2; ThermoElectron, West Palm Beach, FL, USA). Analysis was performed on avian NRPD (AVIAN DB; search terms: chicken, Gallus, Cornix, Aves, turkey, and ostrich; NOT plant, yeast, bacteria, virus) [30].

In addition, a non-avian vertebrate subset of the NRPD (NAVDB; search terms: mammal, *Homo sapiens*, *Rattus*, *Mus*, fish, and excluding the terms used to create the AVIAN DB) was used to identify more proteins. Trypsin digestion was applied in silico to AVIAN DB and NAVDB along with fixed (cysteine carbamido-methylation) and variable (methionine oxidation) modifications. The peptide MS precursor ion mass tolerance was set to 1.5 Da and the fragment ion (MS2) mass tolerance was set to 1.0 Da. Peptide matches were considered genuine if they were greater than 6 amino acids and consistent with described X correlation and ΔCn values. Mass spectral counts were normalized to reduce the sample depth between samples and a likelihood ratio test was performed across leghorn VEX, leghorn TEX and broiler TEX samples to identify significantly different patterns of expression between samples. Proteins with an FDR-adjusted *p-*value (*q* value) of less than 0.05 were considered significant.

Refseq IDs of resulting proteins were mapped against corresponding UniprotKB IDs and unmapped proteins were mapped against UniProt archive (UniParc) to obtain the closest hit. Protein set enrichment analysis was performed to identify proteins previously found in EVs and deposited in the Vesiclepedia database of vesicular proteins (v4.1, 15 Aug 2018) [31]. Functional enrichment of cellular compartment terms was performed using Funrich_V3 [31] .

## 3. Results

### 3.1. Characterization of Exosomes Isolated from Chicken Sera

To compare exosome cargo from the vaccinated and protected, and the challenged and tumor-bearing chickens, serum exosomes were purified from chickens in consecutive vaccine trials (Figure 1A,B). In trial 1, the incidence of MD was somewhat lower in unvaccinated contacts compared to the inoculated shedders, as this trial employed HyLine W36 chickens, which have been bred for MD resistance. In trial 2, however, commercial broiler chickens were used (Ross Yield Plus males X Ross 708 females), which show a more characteristic pattern of MD, in that unvaccinated contacts show an increased susceptibility to MD compared to inoculated shedders (Figure 1B). In Trial 2, which was examining HVT-based vaccines, we used only those chickens having tumors, as we did not have significant protection with these vaccines. Neck-tags for the treatments used in our exosome isolation and characterization are given in Table 1.

To ensure that the particles isolated from serum conformed to exosome size and morphology, we performed nanoparticle tracking analysis of size distribution and concentration of representative samples of serum exosomes (Figure 1C). Particles were 30–150 nm and conformed to the exosome size range with mean sizes of 121, 122.6 and 95.8 nm for VEX, leghorn TEX, and broiler TEX, respectively (Figure 1C). The number of exosomes per 200 μL serum were 6–8 × 10^8^ particles per mL.

To characterize particle morphology, we evaluated the ultrastructure of both leghorn VEX and TEX fractions via negative staining and transmission electron microscopy (TEM) Figure 1D. Our analyses indicated the presence of variable numbers of round and irregularly-shaped vesicles of approximately 50–70 nm in size that displayed exosome morphology (Figure 1D). These vesicles were occasionally observed as dense aggregates and were consistent with our previous observations [22].

### 3.2. Small RNAs Contained within Particles Conform to Exosomal Cargo

We characterized the relative size distribution of RNAs purified from the exosomal particles via bioanalyzer (Figure 2). The RNA yields, size distribution, and profiles were typical of exosome samples, with yields ranging from 0.29–2.27 μg and RNA sizes primarily in the 22–28 nucleotides (nts) range (Figure 2A–C). Additionally, no ribosomal RNAs were evident (Figure 2D).

Results of the deep-sequencing of a total of 12 libraries, composed of quadruplicates leghorn VEX, TEX and broiler TEX samples, are shown in Figure 3 and Table 1. Briefly, we obtained a total of 211,370,657 raw single-end reads with averages of 15.9 million (VEX), 18.6 million (leghorn TEX), and 18.3 million (broiler TEX) reads per treatment group (Table 1). Among the raw reads, about half (112,358,872, 53.2%) passed the quality filters for the presence of 3′ adaptor and insert size of 15–32 nts after adaptor removal (Figure 3A, Table 1). In 10 of 12 libraries, at least 50% of raw reads constituted mappable reads. The peak length distribution of the majority of mappable reads was 21 nts (16%) followed by 22 nts (15.2%) and 20 nts (13.9%), corresponding to the average size range of mature miRNAs (Figure 3B).

Among mappable reads, 27.1% (30,573,661) mapped to the *Gallus gallus* domestic chicken reference genome and 11.7% (13,155,489) mapped to *G. gallus* mRNAs, whereas 7.9% (8,879,834) mapped other RNAs including rRNA, tRNA, snRNA, snoRNA, piwiRNA, etc., currently deposited in the Rfam database. Finally, 485,729 (0.42%) of mappable mRNA reads mapped to the MDV (strain RB-1B) genome.

#### 3.2.1. Cellular miRNAs Compose a Major Key Component of Exosomal Contents

A total of 3055 known unique *G. gallus* miRNAs were detected among the 12 libraries sequenced. Among mappable reads, 30,573,661 (27.2%) reads mapped to known precursor and mature *G. gallus* miRNAs whereas 409,687 (0.36%) reads mapped to known precursor and mature MDV-1 miRNAs (Table 1 and Appendix A).

● miRNAs in Differential Abundance in Leghorn VEX versus TEX

Significant and differentially-expressed (SDE) *G. gallus* miRNAs, reflecting the differential abundance of these species in serum exosomes (greater than 2-fold abundance in leghorn VEX compared to leghorn TEX) included gga-miRs 146b, 143, 10b, 2188, 27b, 99a, 26a, 146c, 24, 146a and 23b (Figure 4, Table 2). Among these, miRNAs, gga-miRs, 146a/b/c share seed sequence but are encoded on different chromosomes (chr 13, 6, and 4, respectively) and were predicted by miRDB algorithm to have common targets [32,33]. On the other hand, gga-miRs, 23b, 27b and 24 are part of a common gene cluster located on the Z chromosome. *G. gallus* miRNAs, miR-30e and -199 were found to be present at nearly equivalent levels in leghorn VEX and TEX, and thus, can serve as reference controls for qRT-PCR normalization (Figure 4, Table 2).

Finally, miRNAs with less than 2-fold abundance difference in leghorn VEX compared to leghorn TEX or conversely, greater than 2-fold abundance in leghorn TEX compared to leghorn VEX include gga-miRs-142, -125b, -181a, *let*-7i, -363, -92, -101, *let*-7g, gga-miRs-451, -148a, -21, -7, -15c, -16, and -16c (Figure 4, Table 2).

Among these, miR-92 and -363 share seed sequence and belong to paralog clusters miR-17-92 (located on chr 1) and miR-106a-363 (located on chr 4), respectively. In addition, *let-7* family miRNAs, *let-7i* and *let-7g* also share seed sequence and are located on chr 1 and 15, respectively. Furthermore, miR-15c, -15 and -16c, also share seed sequence and belong to the miR-15-16 family of miRNAs. The miR-15-16 miRNA family exists in 4 paralog clusters in vertebrates, where the miR-15a/16-1 cluster flanks the TRIM13 gene on chr4, and miR-15c-16c cluster flanks the HPRT gene on chr1, with both clusters sharing a common 7 nt seed sequence.

● miRNAs in Differential Abundance between VEX and Broiler TEX

SDE *G. gallus* miRNAs that were at greater than 2-fold abundance in leghorn VEX compared to broiler TEX included gga-miRs 99a, 2188, 10b, 1388, let-7f, 146b, 26a, 27b, 30e, 146c, and 23b (Table 3). As indicated above, miR-27b and -23b originate from 23b~27b~24-1 cluster and miR-24 was found to be expressed at relatively equivalent levels in both leghorn VEX and broiler TEX. Finally, miRNAs that were less in abundance than 2-fold in leghorn VEX compared to broiler TEX or conversely, those displaying greater than 2-fold expression in broiler TEX include gga-miRs, 199, 425, 181a, 92, let7i, 181a, 126, 148a, 363, 21, 122, 142, 199 and 140 (Table 3).

Those miRNAs expressed in greater abundance in leghorn VEX compared to either source of TEX, were cellular gga-miRs-99a, -2188, -10b, -146b, -26a, -27b, -146c, and -23b. Similarly, miRNAs greater in abundance in either source of TEX compared to VEX include gga-miRs-92, -363, -148a, -181a, and -21, suggesting that these are truly tumor-associated miRs.

#### 3.2.2. MDV-Encoded miRNAs Differ in Abundance between VEX versus TEX

To assess the relative expression of MDV-encoded miRNAs in VEX versus TEX, we examined the relative abundance of the different viral miRNAs species by individual bird numbers and treatments (Figure 5, Appendix A). All MDV miRNAs belonging to the Marek’s disease virus EcoRI-Q-fragment-encoded gene (*meq)*and LAT clusters were detected in VEX and TEX except for miR–M31 (Meq cluster 2), which was identified consistently in broiler TEX and in two samples of leghorn TEX, but not in any VEX samples (Figure 5, Appendix A).

In terms of MDV-encoded miRNAs, we observed the following patterns of expression: (a) the cluster of miRNAs upstream of *meq* were expressed consistently higher in TEX compared to VEX, in particular M4 was present 1–2 orders of magnitude higher in TEX compared VEX (Figure 5, Table 4), (b) the cluster of miRNAs downstream of *meq* were present at lower levels in VEX, and their abundance in TEX was lower than the *meq* upstream cluster (Figure 5C, Table 4), and (c) the LAT cluster of MDV-encoded miRNAs were intermediate in abundance between the *meq* upstream and downstream clusters (Figure 5D, Table 4).

#### 3.2.3. Target Pathways Predicted to Be Affected by VEX and TEX Cellular miRNAs

As we hypothesized that VEX- and TEX-borne miRNAs would target different pathways, we examined the predicted *G. gallus* gene targets for these miRNAs and the number of targets ranged from 6 to 987 for total predictions, and from 8 to 355 for predictions having a gene score ≥80 (Appendix A). To explore the biological processes associated with VEX-upregulated miRNAs, a complete set of miRNA gene targets was used for gene enrichment analysis via DAVID gene ontology (GO) tool [27]. By this analysis, lists of 715 and 901 gene targets were identified for VEX-upregulated miRNAs compared to leghorn and broiler TEX, respectively (Appendix A). DAVID successfully mapped 690 of 715 and 864 of 901 gene symbols to *G. gallus* genes.

KEGG pathway analysis predicted 6 pathways likely to be affected by VEX-borne miRNAs, and 13 and 10 pathways likely to be affected by leghorn and broiler TEX miRNAs, respectively (Appendix A). The pathway predicted to be most significantly affected by VEX was the mitogen-associated protein kinase (MAP kinase) pathways (MEK/ERK, p38 and JNK) (Appendix A). This was the only pathway to have a false-discovery rate (FDR) <0.05.

In terms of pathways most likely to be affected by both leghorn and broiler TEX, the phosphatidylinositol signaling pathway had false-discovery rates (FDR) ≤0.05 for both sets of TEX (Appendix A). Among the enriched pathways, the top 8 pathways predicted to be affected were the same between TEX isolated from layers and broilers, albeit not in the same order (Appendix A). Of the pathways predicted to be affected by TEX, five were also predicted to be targeted by VEX, as well, although different members of the pathways were the predicted targets (Appendix A).

#### 3.2.4. VEX and TEX Contain MDV mRNAs

In addition to miRNAs mapping to the genome, we identified mRNAs which also mapped to the MDV genome (pRB-1B reference strain) (Figure 6). The number of reads that mapped to reference sequence to produce a contig were 86,000 (leghorn VEX), 122,113 (leghorn TEX) and 265,616 (broiler TEX) (Table 1). Since the unique mappable reads obtained in our current study were the result of size selection prior to small RNA sequencing, and were further subjected to quality control steps to ensure read lengths between 15–32 nts, reads mapped to the reference genome may be the result of sequencing of fragmented exosomal transcripts or incomplete reverse transcription products during sample preparation. 

The number of raw reads accounted for mapping to the coding regions of MDV genome were 133,627 (leghorn VEX), 1242 (leghorn TEX), and 1044 (broiler TEX) (Appendix A). A total of 140 genes (minimum of 5 reads/transcript) were identified in sample libraries corresponding to leghorn VEX, whereas only 6 and 16 genes had at least 5 reads mapped in sample libraries corresponding to leghorn TEX and broiler TEX, respectively (Figure 6, Appendix A). The lack of raw read counts and normalized expression values for a major proportion of MDV genes in leghorn or broiler TEX precluded us to evaluate differential gene expression between leghorn VEX and leghorn or broiler TEX.

Upon normalizing raw read counts, gene expression values indicated the top 20 genes of highest expression in their descending order of expression including UL34 (nuclear envelopment in complex with UL31), RLORF12, UL35, UL42 (viral DNA polymerase processivity factor), UL20, RLORF7 (Meq), UL33 (minor capsid protein), MDV103, LORF9, UL40 (ribonucleotide reductase), UL29 (single-stranded DNA binding protein), UL9 (origin binding protein), UL32 (procapsid chaperone and viral oxidoreductase), L1, UL46 (VP11/12 tegument protein), US6 (glycoprotein D), UL19 (major capsid protein), UL24, RLORF13 (pp38) and UL23 (viral thymidine kinase) (Figure 6, Appendix A) [34,35].

Among these were genes that have essential functions in herpesviral genome replication (UL9, UL42, U29, UL40, UL23), capsid assembly (UL33, UL35, UL19), DNA encapsidation (UL32), nuclear egress (UL34), cytoplasmic capsid maturation, and egress (VP11/12, US6, UL24, UL20) [24,29,35,36].

In addition to these, normalized abundance analysis indicated gene expression from a majority of MDV genome in VEX, including gene segments from both unique and repeat regions (Figure 6, Appendix A).

In contrast to VEX, normalized read abundance in leghorn TEX mapped to only [4] genes including RLORF8, L1, ICP4, and UL30 in descending order of expression. Other gene products not described here, had read counts were less than 5 (Figure 6, Appendix A). The remaining MDV gene products lacked any expression in leghorn TEX.

In broiler TEX, reads corresponding to a total of 14 MDV genes were identified as having read numbers >5 including RLORF8, L1, UL3, UL34, LORF3, RLORF1 (viral telomerase RNA or vTR), UL12, UL46 (VP11/12), UL32, ICP4, UL8 (DNA helicase/primase complex associated protein), UL17 (tegument protein), and UL36 major tegument protein (MTP) (Figure 6, Appendix A).

Of note, the MDV *meq* cluster 1 miRNAs, miR–M4, and –M2, whose read counts accounted for 66.73% and 1.76% of total number of miRNA reads, respectively, in leghorn TEX, and 36.35% and 6.36% of total number of miRNA reads in broiler TEX, respectively, fall within the RLORF8 coding region (Appendix A). Thus, the greater expression of MDV-miR-M4 in leghorn or broiler TEX may account for the highest expression of RLORF8 as precursor RNAs. In general, leghorn and broiler TEX mRNAs mapping to the MDV genome were in much lower abundance than those contained in VEX (Figure 6, Appendix A), with the exception of those mapping to the repeats flanking the unique long region (IR_L_/TR_L_).

### 3.3. Determining Protein Biomarkers of VEX and TEX

To assess differential protein content between VEX and TEX, an average 358 ± 48 proteins (mean of 1386 ± 86 peptides) for VEX, an average of 461 ± 80 proteins were identified (mean of 1643 ± 354 peptides) for leghorn TEX, and 420 ± 30 proteins (mean of 1739 ± 129 peptides) were identified from broiler TEX (Appendix A). Of these, 69 proteins were common to all three exosome preparations, and 221 and 208 proteins were unique to leghorn and broiler TEX, respectively, suggesting that cellular transformation by MDV significantly altered exosome content (Figure 7). Of the identified proteins, 213/317 (67%), 309/482 (64%), and 292/433 (67%) were successfully mapped by Funrich functional enrichment tool among which, 193, 282, and 258 proteins were present in the Exocarta database of previously reported exosome proteomes (Figure 7B, Appendix A).

The unmapped proteins also included previously identified exosomal proteins, for instance, histone subunits (H2A, H2B), integrin subunits (ITGB), actin subunits (ACTA3, ACTSG, ACT5, ACTC), tubulin chains (TBA2, TBA3, TBA4A, TBB2, TBB4, TBB5, TBB6), cytochrome subunit (CYTB), alpha-2-microglobulin –like 4 (A2ML4), cytokine receptor chain subunits (IL10R1, gp130) and MDV proteins, which remained unrecognized by Funrich tool due to variations between *G. gallus* nomenclature and respective orthologs belonging to *H. sapiens*, which is used by Funrich (Appendix A). These data further suggest that the vesicle preparations in our study contained abundant levels of exosomes. 

The proteomic analysis also identified 71 novel exosome proteins not present in *Exocarta*, as indicated in the Venn diagram (Figure 7B). As anticipated with the latent and tumorigenic phase of MDV infection, there was a significant alteration in exosome content, of which 45 proteins common to both leghorn and broiler TEX were not identified in leghorn VEX (Figure 7B, Appendix A).

By convention, exosome characterization relies on the detection of common marker proteins enriched in exosome fractions. These include tetraspanins (CD9, CD63, CD81), exosome biogenesis proteins (Tsg101, Alix), and others (Syntenin-1, Flotillin-1) [38]. However, none of these except tetraspanin-10 were detected in our exosome samples. Nonetheless, exosomal proteins previously reported to be associated with vesicle secretion in cancer cell line supernatants (RAP1A, RAP1B, VAMP1, MYH7, MYO18a, MYOLPF, MYO1E, VPS13B, HSP70) were identified in our samples [38]. Furthermore, comparison to the Vesiclepedia database revealed 32.8–41.5% of the proteins identified in our datasets were predicted to be exosomal, and the majority were localized to endolysosomal compartments (Figure 7C).

Exosomal protein functional groups in leghorn VEX include membrane receptors (C1r, EphA4), metabolic enzymes (GAPDH, LCAT, PCY1OXL), heat shock proteins (HSPA9), cytoskeletal proteins (ActA, ActB, ActG1, myo1E, tub2, -4, -5, -7), histones (H2A-I, -II, -IV, VI, H2B), ribosomal proteins (mRPS31), vesicle trafficking proteins (VAMP1, STXBP4, ArfGef12). The only MDV-encoded proteins identified in low abundance were immediate early protein, ICP4 in VEX, and UL36 (major tegument protein), UL47 (tegument protein) in TEX of both origins (leghorn and broiler) (Appendix A).

Among overrepresented proteins identified in leghorn VEX in comparison to both TEX samples include collagen α-1, XXII chain (COL22A1), and IGFBP-complex acid labile subunit isoform (IGFBPALS) (Appendix A). COL22A1 is associated with tissue integrity, myotendinous junctions, and cellular adhesion. IGFBPALS, aka IGFBP-5, is similarly associated with systemic tissue integrity, and has been previously demonstrated to be expressed in exosomes [38].

Common functional groups of exosomal proteins identified in leghorn TEX included β_2_m, membrane receptors (THRB, TSPAN10, C1r), metabolic enzymes (PKM, ENO1, GAPDH, LCAT), elongation factors (Eef1a2, Eef1B2, Eef2, Eif2s3, eIF4A2), heat shock proteins (HSPA4L, Hspbp1), cytoskeletal proteins (ActA, ActB, ActG2, myl6, my015a, tubb, tuba, DNAH), histone subunits (H3-I, -II, -III, -IV, -V, -VI, -IX, H4-I, -II, -III, -IV, -V, -VI, H2A-VII, H2A-VI, H2A-VIII), ribosomal proteins (RPL-4, -5,-6,-10a,-17, -P1, -P2, RPS-2, -6, -19), vesicle trafficking proteins (VPS13A), chromatin modifying enzymes (KAT5, NSD1, H3K36me, H4K20me, ATM) and pro-apoptotic protein (BID) (Appendix A). 

Common functional groups of exosomal proteins identified in broiler TEX include β_2_m, membrane proteins (CD36, CD48, CD5L, CD163L1), membrane receptors (INSR, CFTR, DMBT1), metabolic enzymes (PKM, ENO1, GAPDH, LCAT, PCYOX1L), elongation factors (Eef1a2, Eef2S3), heat shock proteins (Hsp5), cytoskeletal proteins (ActA1, ActB, ActA2, ActBl2, ActG1, myh6, myh7, tubb, tuba), histone subunits (H2B-I, H2B-II, H2B-III, H2B-IV, H2B-VI, H2A-VII, H2A-VI, H2A-VIII), ribosomal proteins (RPL-4, -5, -6, -10a, -17, -P1, P2, RPS-2, -6, -19), vesicle trafficking proteins (RanBP), and transcription factors (JunD, E2F4, YEATS2) (Appendix A). 

#### 3.3.1. Enrichment Analysis of Exosomal Transcription Factors (TFs)

The presence of viral and cellular transcription factors (TF) in leghorn VEX or TEX (leghorn and broiler) permitted gene enrichment analysis based on transcription factors and associated targets in proteomic datasets (Appendix A). The top 6 transcription factor networks enriched in each treatment group were identified. While *Bonferroni*-corrected TF networks pertaining to HNF1A and ARID3A remained common to leghorn VEX and leghorn and/or broiler TEX, respectively, TF networks FOXA1, RXR, RREB1 and YY1 remained unique to leghorn VEX (Appendix A, Appendix A). Similarly, *Bonferroni*-corrected TF networks associated with Jun, FosB and NKX6.1 remained common to leghorn and broiler TEX (Appendix A). The only TF network unique to leghorn TEX was MEF2A, whereas those networks uniquely-identified to broiler TEX include ONECUT1 and Fos (Appendix A).

#### 3.3.2. Putative Protein Biomarkers of VEX and TEX

Comparison of proteins consistently-identified in VEX or TEX (leghorn or broiler) identified several proteins that were indicative biomarkers of VEX or TEX (Table 5; Table 6, Appendix A). In VEX of leghorn chickens, collagen α1 XXII (COL22A1) was identified as significantly more abundant compared to TEX of layers (>10,000-fold, *p* < 0.05), and broilers (>25-fold, *p* < 0.05) (Table 5). Similarly, insulin-like growth factor, acid-labile subunit (IGFALS) was more abundant in VEX than in TEX of leghorns (>1000-fold, *p* < 0.02), and broilers (>850-fold, *p* < 0.05). Consequently, each of these may serve as exosomal biomarkers for assessing vaccine-induced protection from MD. 

There were additional proteins identified consistently-associated with protection (protocadherin-17, IgGFc-binding protein-like protein, tenascin receptor, Muellerian-inhibiting factor, transferrin receptor, and the C-reactive protein, pentraxin-related precursor, etc., (see Table 5); however, these differences in detection were statistically significant in one comparison (i.e., leghorn VEX vs. TEX) but not the other (Appendix A).

In terms of TEX-associated marker proteins, isoforms of pantetheinase (aka, vanin-1) were consistently identified in TEX, but not in VEX (leghorn TEX > 12,000-fold, *p <* 0.01; broiler TEX > 6500-fold, *p* < 0.02), see Table 6. Additional markers included histone H4, α-2-antiplasmin isoform X2, HSP90-α, peroxiredoxin-1, deleted in malignant brain tumors-1 protein-like isoform X2, ceruloplasmin, clusterin, and mannose-binding lectin precursor, etc.; however, these were only statistically significant in one comparison (e.g., leghorn TEX vs. VEX), but not the other (Appendix A). 

## 4. Discussion

This is the first report of the complete transcriptomic and proteomic analyses of chicken serum exosomes from commercial leghorn and broiler Marek’s disease virus vaccine studies. The cellular cargo miRNAs showed distinct patterns of abundance suggesting that VEX miRNAs target the MAP kinase, cellular proliferation pathway (Appendix A). Both sets of TEX-borne miRNAs showed significant targeting of phosphoinositol signaling (Appendix A), suggesting that these would disrupt pathways essential to lymphocyte activation, and would therefore likely be immunosuppressive.

### 4.1. Abundant VEX-Borne miRNAs

As mentioned in the results, the most abundant miRNAs found in VEX were the miR-146 family, the miR-23 cluster, and miR-26a, among others. These miRNAs have been associated with modulation of innate immune responses and T-cell patterning, and are described briefly, below.

#### 4.1.1. The miR-146 Family

MiR-146a and -146b share seed sequences, and thus have common predicted targets and differ by two 3′-end nucleotides. These were initially identified as endotoxin-inducible miRNAs via TLR4 signaling [32,33]. miR-146a and -146b function as negative regulators of NFκB, by targeting IRAK1 and TRAF6, resulting in decreased pro-inflammatory cytokine production. In our study, chicken IκB kinase subunit β (IKBIP or IKKβ), was found to be a high confidence target (miRDB score: 84) of miR-146a/b, suggesting that VEX may negatively regulate NF-κB.

#### 4.1.2. The miR-23~27~24 Cluster

This cluster consists of 3 members existing as two paralogs (a and b) [39]. Mature miR-23a and miR-27a differ by one nucleotide with respect to their paralogs miR-23b and miR-27b, whereas miR-24-1 and -2 share the same sequence. Although it is hypothesized that evolutionarily-conserved miRNA clusters cooperatively target the same gene or various components of a common biological process, members of 23~27~24 cluster have antagonistic roles rather than cooperative roles in modulating effector T-cell differentiation [39].

Cellular miR-23b displayed anti-inflammatory properties in human cells by directly repressing NFκB signaling activators, TGF-β activated kinase 1/MAP3K7 binding protein 2 (Tab2), Tab3, and inhibitor of nuclear factor B kinase subunit α (IKK-α) [40]. Among the above, we identified Tab3 (miRDB score: 95) and a previously confirmed target gene chicken IRF1 (miRDB score: 91) as miR-23b predicted targets, corroborating convergent evolution and suggesting that miR-23b may also contribute to pro-inflammatory cytokine downregulation.

Cellular miR-24 overexpression was demonstrated to promote Th1 and Th17 differentiation of naive T cells [39]. In addition, miR-24 can also target IL-4 expression in an indirect manner as shown for miR-27 mediated targeting of GATA3 by targeting Bmi1, a molecule that stabilizes GATA3 by blocking its proteasome-dependent degradation [41]. The observed greater levels of miR-27b and -24 compared to miR-23b in VEX, suggest that MD vaccination leads to maintained Th1 differentiation.

A putative tumor suppressor role for miR-27b is suggested by its deletion in a docetaxel-resistant luminal-type human breast cancer cell line [42]. Similarly, in numerous cancers, forced-expression induces a loss of invasiveness in numerous cancer cell lines [43,44]. In this regard, several of the miR-27b targets: LIMK1 (miRDB score:98), CDH5 (miRDB score:97) and VEGFC (miRDB score:62) were among predicted chicken targets of *G. gallus* miR-27b.

Cellular miR-24, conversely, displays a context-dependent role as a tumor suppressor or pro-oncogenic miRNA, in that it suppressed cell proliferation and migration in prostate cancer, arterial smooth muscle cells, vascular muscle cells, nasopharyngeal carcinoma, bladder cancer, and small-cell lung cancer, but acts as an oncomiR in glioma, Hodgkin’s lymphoma, and SQCC [45,46]. In the context of MD lymphomas, the relative high abundance level of miR-24 in VEX suggests that it may function in a possible tumor suppressor role.

#### 4.1.3. miR-26a

Cellular miR-26a functions as a tumor suppressor miRNA in nasopharyngeal carcinoma (NPC), breast cancer, lung cancer, and hepatocellular carcinoma (HCC), although it has also been associated with tumor cell proliferation in glioma and CCA [42,47,48,49]. Additional previously-validated miR-26a targets identified in our prediction analysis include SMAD1, MTDH, CDK6, PTEN, and MAP3K2 (Figure 6A) [50,51]. A further suggestion of the tumor suppressor role of miR-26a in the context of MD lymphomas was the finding that miR-26a is significantly-downregulated in vMDV (GA)-infected spleen and liver tumors compared to uninfected spleens and uninfected PBLs, respectively [52]. miR-26a was found to be downregulated in seven independently-derived MDV-transformed lymphoblastoid cell lines, as well as ALV-and REV-transformed lymphoblastoid cell lines [53].

#### 4.1.4. VEX-Borne miRNA Targeted Cellular Pathways

In terms of VEX-borne miRNAs, our pathway prediction suggests that mitogen-associated protein kinase (MAPK) signaling is targeted for downregulation at multiple targets (Appendix A).

### 4.2. TEX-Borne miRNAs

The most abundant miRNAs consistently found in TEX were miR-92/-363 cluster, let 7i and 7g, and miR-21. These miRNAs, as well as MDV-encoded miRNAs, have been associated with various malignancies, and their abundance in TEX suggest contributory roles in MDV-mediated transformation and immune suppression.

#### 4.2.1. miR-92 and -363

Cellular miR-92 is transcribed as part of a polycistrionic miR-17-92 cluster, a.k.a oncomiR-1, which encodes 6 miRNAs including miR-17, -18a, -19a, -19b-1, -20a and -92a [54]. In addition to the miR-17-92 cluster, there exist two paralog clusters including the miR-106a-363 cluster (containing miR-106a, -18b, -20b, -19b-2, -92-2 and -363), and the miR-106b-25 cluster (containing miR-106b, -93, -25) which encodes miR-92 paralogs miR-92a-2, -363 and -25, respectively [55]. Although miR-92a-2 and miR-363 are co-expressed as part of polycistrionic transcript, varying post-transcriptional processing results in differential processing of these mature miRNAs [55].

In TEX, miR-92 and -363 were found to be in greater abundance compared to VEX. In addition, with a read number less than 2000 (below the set detection limit) remaining miRNAs of the miR-17-92 cluster (miR-17, -19a, -20a), and miR-106a-363 cluster (miR-20b, -106) were also found in greater abundance compared to VEX. 

Amplification of the miR-17-92 cluster-coding region is found in cell lines representative of several types of hemopoietic malignancies including: splenic lymphoma with villous lymphocytes, six mantle lymphoma cell lines, ATN-1 (adult T-cell lymphoma cell line), Jurkat (T-cell acute lymphocytic leukemia), HT-1 (T-cell lymphoblastic lymphoma cell line, HTLV-1-negative) and HUT78 (peripheral T-cell lymphoma cell line) [56]. MiRs -19a, -20 and -92a were overexpressed in these lymphoma cell lines, whereas miR-17a and -18 were found at lower levels. Furthermore, overexpression of the miR-17-92 cluster in transgenic mice caused B cell lymphomas [57]. Based on greater expression of miR-92 and -363 in TEX and their previous roles as oncomiRs, our data suggest that they may contribute to MDV-mediated lymphomagenesis.

#### 4.2.2. *Let*-7i and -7g 

The cellular miRNAs *let-7g* and *let-7i* belong to the *let-7* family of miRNAs, which are known tumor suppressor miRNAs downregulated in a variety of malignancies. In contrast to their role in many cancers, let-7i was found to be highly upregulated in follicular lymphoma tissues from patients that underwent transformation (FCL-t). Likewise, let-7a, -7b, -7i and -7f were consistently upregulated in Hodgkin’s lymphoma cell lines and tissues irrespective of EBV-infection status [58].

#### 4.2.3. miR-21 

Cellular miRNA mir-21 is a *bona fide* oncomir that is overexpressed in virtually all carcinomas and various hematological malignancies including MDV1-induced lymphomas, and is transactivated by Meq [59,60]. Mir-21 targets numerous gene products including MyD88 and IRAK1 to promote immune suppression [61].

#### 4.2.4. MDV-Encoded miRNAs

Other TEX-borne miRNAs with potential roles in MD lymphomagenesis include the MDV miRNA clusters. In terms of MDV-encoded miRNAs, we found that most were identified in both VEX and TEX; however, the *meq*-cluster 1 (cluster upstream of *meq*), were the most abundant species identified and were of ~10-fold greater abundance in TEX compared to VEX (Figure 5). MDV-encoded miRNAs downstream of *meq* were in least abundance (~100-fold less than the upstream cluster), with significant bird-to-bird variation in the abundance of the different species; however, in leghorn TEX, this cluster was more consistently-identified than in VEX. The latency-associated transcript (LAT) cluster of miRNAs were intermediate in expression between the *meq*-upstream, and -downstream cluster miRNAs. This LAT cluster was identified in 3 to 4-fold greater abundance in TEX than in VEX.

Overall, our data support the role of the *meq*-upstream miRNA cluster, particularly M4, in contributing to systemic immune suppression and likely contributing to cellular transformation [5,62,63].

#### 4.2.5. TEX-Borne miRNA Targeted Cellular Pathways

In our studies of TEX-borne cellular and viral miRNAs, we found that the consistently-targeted cellular genes by TEX from either source, were those involved in phosphoinositol signaling. Targeted genes in this signaling includes class I PI3K catalytic subunits (PIK3CB/PI3Kβ, PIK3CD/PI3Kδ), PI3K regulatory subunit (PIK3R1/p85α), downstream PI(3,4,5)P_3_ phosphatase (PTEN), PI3P kinase (PIKfyve), phosphoinositide hydrolases (PLCB, PLCB1, PLCL2), diacylglycerol (DAG) kinases (DGKB, DGKD, DGKH), inositide triphosphate (IP_3_) receptor (ITP1R) and inositol phosphatases (IMPA1, IMPAD, INPP5A, SYNJ1, SYNJ2, MTMR6, MTMR12) (Appendix A, Appendix A, Appendix A). 

Class I PI3K, PI3Kδ mediates downstream signaling of growth factors, cytokines, and immune receptors such as B-cell, T-cell, and Toll-like receptors [64]. The targeting of upstream catalytic and regulatory subunits to prevent downstream signaling may promote TEX-mediated immune suppression. On the other hand, PTEN balances and counteracts PI3K activity by PI(3,4,5)P_3_ hydrolysis, and attenuated expression, loss of function mutations, or total loss of PTEN is indicated in many forms of malignancies, while PIK3CA over-activation, due to mutations, is also indicated in cancer progression [64].

### 4.3. VEX/TEX-Borne MDV mRNA Expression

In terms of the MDV-encoded mRNAs identified in VEX and TEX, we report a very stark, and likely incredibly important difference in the abundance of transcripts between these exosomes. Contrary to our initial hypothesis, in which we believed that MDV-associated mRNAs would be primarily identified in TEX, and would map exclusively to the transformation- associated genes (*meq*, RLORF4, vTR region), we found much higher levels of MDV mRNAs in VEX, and moreover that these map across the entire MDV genome, spanning virus-encoded structural, enzymatic, and regulatory genes (Figure 6).

As vaccine MDVs typically replicate early (1–2 weeks post-vaccination), and then establish low levels of latent infection with intermittent periods of virus replication, our data suggest a paradigm shift in how MD vaccines elicit long-term anti-viral immunity. Our data suggest that vaccine (and/or perhaps challenge virus-infected cells) are shedding functional viral mRNAs packaged in serum VEX, and that these may then fuse with antigen presenting cells (APCs) throughout the body, presenting the mRNA-encoded peptides in the context of MHC-I, stimulating continual CTL activation in the absence of overt viremia.

Interestingly, we observed very few MDV proteins in VEX and TEX, save ICP4 (VEX) and tegument proteins (TEX), described below (Appendix A), suggesting that the main mechanism of cross-presentation of these antigens to immune cells may be in the functional mRNA form. Our preliminary in vitro findings indicate exosomal-transfer of functional MDV mRNAs to recipient chicken macrophages, and we are further validating our findings by treatment of chicken macrophages with VEX and TEX for the detection of protein expression. The process of exosomal-transfer of mRNAs to cells in which they become translated has been shown previously, and has been associated with the maturation state of APCs with which they fuse [65], suggesting that mature dendritic cells have the greatest affinity and expression of these proteins.

### 4.4. Proteomic Profiles of VEX and TEX 

#### 4.4.1. Potential VEX and TEX Biomarkers 

In terms of our proteomic analysis of VEX and TEX, we identified common and exosome source-specific proteins (Table 5 and Table 6 and Appendix A). In terms of VEX-specific proteins, we identified a number of proteins with putative effects in the regulation of immune responses.

The most consistently identified proteins in this source of VEX were IGFALS and COL221A, suggesting that they may serve as biomarkers of vaccine-induced protection. Likewise, among overrepresented proteins identified in TEX, pantetheinase, encoded by the vanin-1 gene, can serve as a possible serum biomarker for MDV tumor development. Vanin-1 is an enzyme involved in catabolizing intermediate pantetheine to vitamin B5 for recycling in coenzyme A biosynthesis [66]. Vanin-1 contains a GPI anchor sequence, is membrane bound on epithelial and myeloid cells, and is a potential biomarker for certain inflammatory conditions, such as pancreatic cancer associated diabetes (PCAND) and renal injury [45]. By regulating the synthesis of cysteamine and glutathione, vanin-1 affects cellular viability, proliferation, and function [67]. In PCAND, neoplastic cells overexpressing vanin-1 displayed greater paraneoplastic islet injury through an increased oxidative stress response. Based on our data, we propose exosomal pantetheinase may serve as a serum biomarker for MDV-induced tumor development.

#### 4.4.2. TEX Contain MDV Tegument Proteins

Among MDV proteins identified in TEX, the major tegument protein (MTP) encoded by UL36 (MDV049) gene was detected in TEX of either source (leghorn or broiler) (Appendix A). MTP is expressed both during lytic and latent infection and encodes an ubiquitin specific protease (USP) in its N terminus that is conserved among all known herpesviruses [6,68]. A C98A, protease-dead mutation, in its USP catalytic site, leads to severe reduction in MD tumor incidence, although MDV replication was unaffected in vitro and in vivo. HSV1 USP was shown to deubiquitinate poly-ubiquitinated IκBα and mono-ubiquitinated PCNA to inhibit HSV1 DNA-induced IFN-β or NF-κB activation and DNA damage responses, respectively, to facilitate infection [69,70].

In addition, tegument protein VP13/14 encoded by UL47 was exclusively detected in leghorn TEX (Appendix A). UL47 is a late gene expressed during lytic infection with very low and high abundances in CEFs and FFE, respectively [24]. Origins of UL47 in TEX may be due to virus reactivation from latency and selective exosomal packaging for transfer to distant FFE in order to promote transmission. 

#### 4.4.3. VEX Contain MDV ICP4

Interestingly, the IE transactivator ICP4 was found in low abundances in VEX, but not in TEX. Syngenic cell-mediated immune responses towards ICP4 was found to be a major factor in defining the genetic resistance towards MD in B^21^B^21^ chickens [9]. Exosomal transfer of ICP4 to APCs and the induction of subsequent CTL responses towards conserved ICP4 epitopes appears to be a critical mechanism in inducing protection against MDV.

#### 4.4.4. Transcription Factor-Associated Pathways in VEX and TEX

Since we found that numerous transcription factors, histones, and DNA-modifying enzymes were present in both VEX and TEX, we developed models to provide some insight into the potential effects of these factors on recipient cells. Transcription factor-associated pathway modeling suggested that there were some TF networks common to VEX and TEX (hepatocyte nuclear factor, HNF1A and A+T-rich interaction domain, ARID3A, Appendix A, Appendix A), which play important roles in mouse early embryonic development and maturation of tissues (liver, pancreas and kidney) and hemopoietic stem cell (HSC) progenitors, respectively [71,72].

VEX-specific transcriptions factors included RREB1, RXRA, and YY1 (Appendix A, Appendix A). While RREB1 is a proto-oncogene and transcription factor upregulated in several solid tumor malignancies, but its role in MD lymphoma formation remains unknown [73]. RXRA is a retinoic acid superfamily nuclear receptor that homo- or heterodimerizes with other nuclear receptors to mediate a wide range of actions in myeloid cells including chemokine expression, anti-viral Type I IFN attenuation, phagocytosis of apoptotic cells, and finally, repression of metastasis promoting genes, suggesting similar roles in MD vaccine mediated protection [74,75].

YY1 is a GLI-kruppel family transcription factor that can activate or deactivate gene expression depending on binding partners and has a context dependent role as a tumor suppressor or a tumor growth promoter [76]. In MDV-transformed CD30^hi^ cells, YY1 protein expression was significantly downregulated compared to CD30^lo^ cells indicating a possible tumor suppressant role [7,76].

Unique to TEX, the MEF2A TF network was identified suggesting a role similar to previously identified roles of MEF2 family TFs, MEF2A, MEF2B and MEF2D in EBV latency maintenance and reactivation. MEF2 TFs, through interaction with MEF2 binding motifs in the immediate early (Zta, Rta) gene promoters maintain them in a silenced state by recruitment of class II histone deacetylases (HDACs), whereas upon HDAC inhibitor treatment, Zta expression and EBV reactivation results from MEF2B/D recruitment of histone acetyl transferases and active chromatin modifications of Zta promoter [77,78,79].

MEF2 TFs were reported to be expressed in MDV-transformed CD30^hi^ cells in a previous proteomics study reported by Shack, *et al.*, [7]. Furthermore, in leghorn or broiler TEX, the AP1 family TF networks involving Jun, Fos, and FosB, and TFs (JunD) themselves were identified, and these are known to heterodimerize with the MDV oncoprotein Meq to promote cellular transformation [80,81].

## 5. Main Conclusions

The most compelling finding of our work is the identification of MDV-encoded mRNAs spanning the entire genome in VEX, at a time when circulating vaccine virus is not readily detectable in PBMC, suggests a novel mechanism by which CTLs can be continually primed systemically, through fusion of APCs with VEX and MDV antigen presentation. This report also provides a number of important data sets for follow-up studies on TEX in Marek’s disease virus-mediated immune suppression and lymphomagenesis, as well as provides putative mechanisms by which vaccine-associated exosomes (VEX) may lead to anti-proliferative responses and prolonged, systemic protection from MDV-induced lymphoma formation.

## Figures and Tables

**Figure 1 genes-10-00116-f001:**
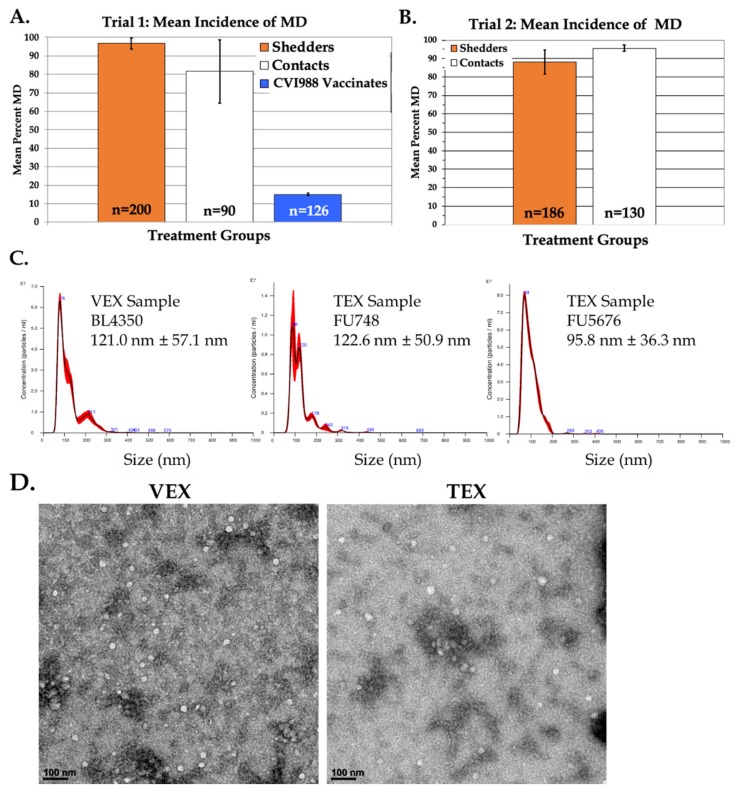
Exosome source, size, concentration, and ultrastructural characterization. Panels (**A**,**B**) show the Marek’s disease (MD) incidences of inoculated shedder (orange), contact-exposed, unvaccinated (white), and CVI988-vaccinated (blue) treatment groups. The numbers of birds for each treatment are given in the bars. Panel (**C**) shows the nanoparticle size distribution (± SD) and particle concentrations (log particles/mL) of representative VEX and TEX samples. Panel (**D**) shows representative transmission electron micrographs of Total Exosome Isolation (TEI) reagent-purified VEX and TEX. Vesicles were typically round and varied in diameter from approximately 30–120 nm. Scale bar = 100 nm.

**Figure 2 genes-10-00116-f002:**
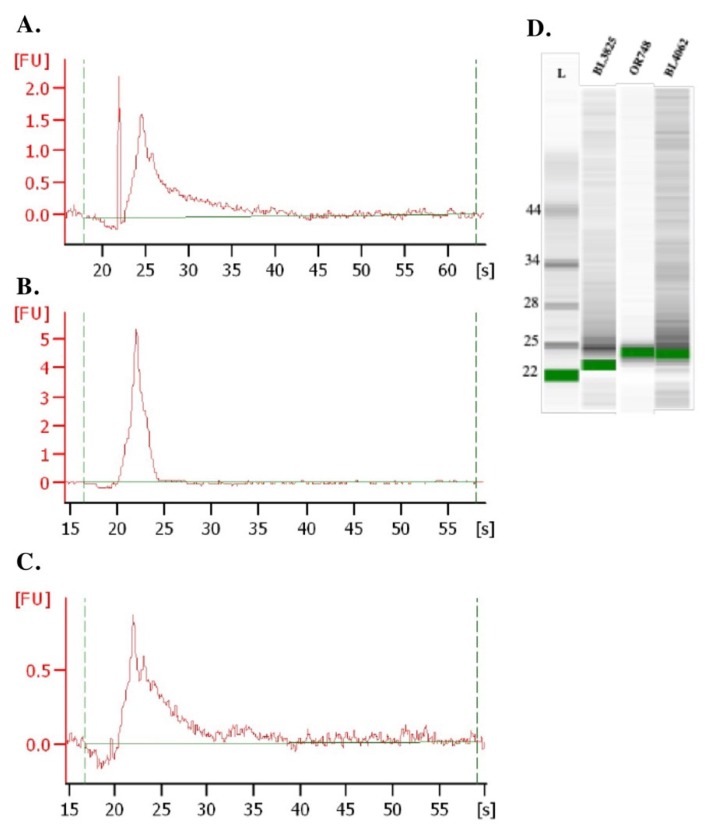
Size distribution of exosomal RNAs. Panels (**A**–**C**) show fluorograms of exosomal RNAs determined by the Agilent RNA Pico Chip. Panel (**A**) shows fluorogram of exosomal RNA from leghorn VEX (Bird tag# BL3825), leghorn TEX (Bird tag# OR748) (**B**), and broiler TEX (Bird tag# BL4062) (**C**). Panel (**D**) shows RNA chip analyses of purified exosomal RNAs.

**Figure 3 genes-10-00116-f003:**
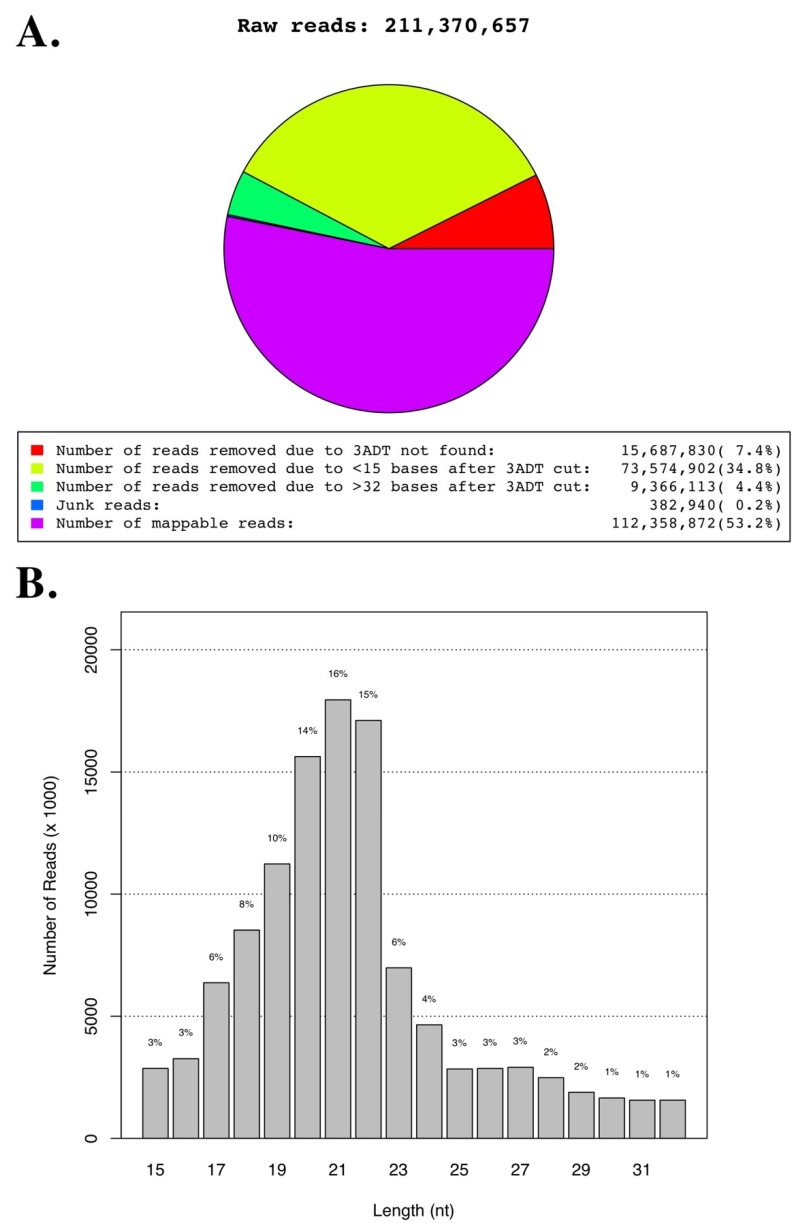
Read statistics of exosomal RNAs. Panel (**A**) shows a pie chart of the read statistics of the small RNA libraries used for VEX and TEX transcriptomic analyses. Panel (**B**) shows the read lengths of the small RNA transcriptome libraries.

**Figure 4 genes-10-00116-f004:**
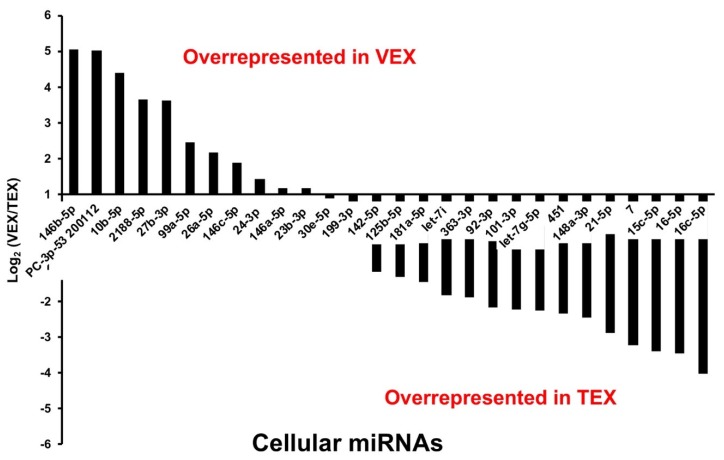
Exosomal cellular miRNAs of different abundance in VEX and TEX. The bar graph above shows a comparison of the Log_2_ (PVL/TBL) of miRNA mean read number, where PVL are VEX from vaccinated and protected leghorns, and TBL are TEX from inoculated shedder leghorns. Those to the left are significantly over-represented in VEX, those to the right are significantly over-represented in TEX (*p* < 0.01).

**Figure 5 genes-10-00116-f005:**
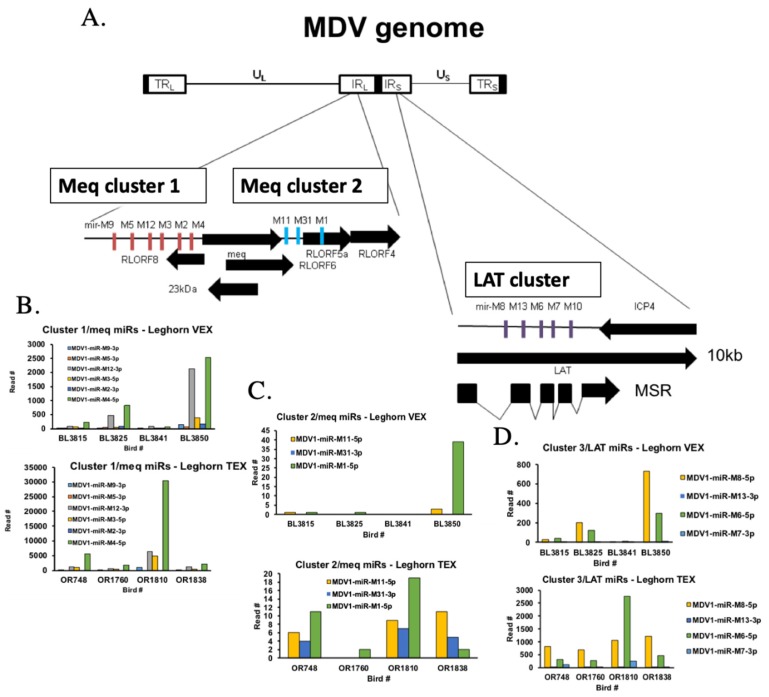
Comparison of MDV-encoded miRNAs in VEX and TEX. Panel (**A**) shows a map of the MDV genome with the location and orientation of the miRNA clusters (Meq cluster 1, 2 and LAT cluster). Panel (**B**) shows the read numbers of the Meq cluster 1 miRNAs in leghorn VEX (top) and TEX (bottom) showing reads for individual chickens. Note differences in y-axes (3000 vs. 35,000). Panel (**C**) shows the Meq cluster 2 (downstream of the *meq* gene) miRNA reads for leghorn VEX (top) and TEX (bottom). Panel (**D**) shows the individual reads of the LAT cluster of miRNAs from leghorn VEX (top) and TEX (bottom).

**Figure 6 genes-10-00116-f006:**
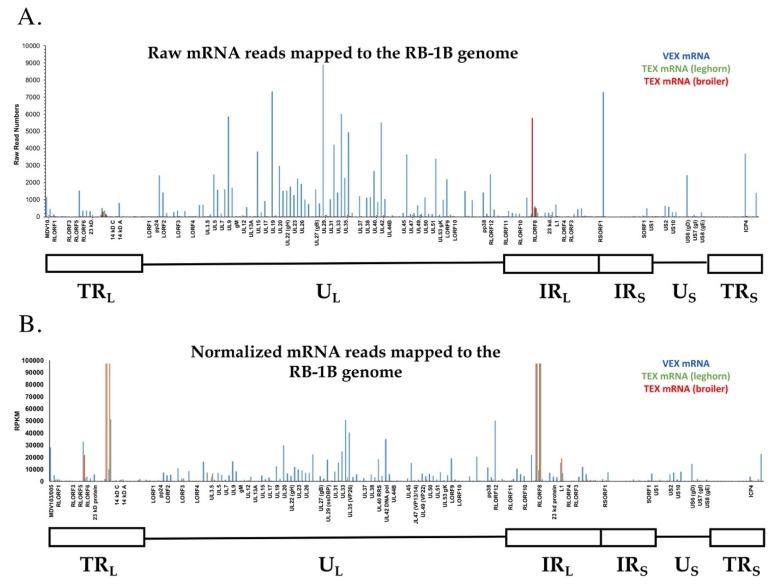
Exosomal mRNAs mapping the MDV genome. Panel (**A**) shows the raw read number of exosomal mRNAs mapping to the MDV genome from leghorn VEX (blue), leghorn TEX (green) and broiler TEX (red). Panel (**B**) shows these data normalized to gene size and read abundance (RPKM). Note localization differences between VEX (numerous MDV structural, enzymatic and regulatory genes), and TEX (primarily latency/transformation-associated genes). For reference, common gene names and maps of the MDV genome are included along and below the *X*-axis, respectively, in each panel.

**Figure 7 genes-10-00116-f007:**
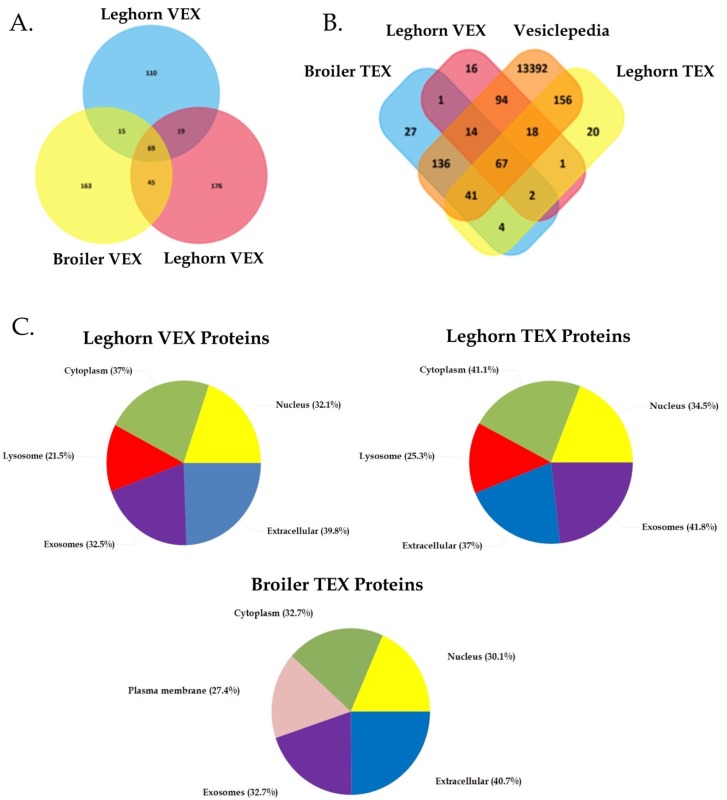
Proteomic analysis of leghorn VEX, TEX and broiler TEX. Pane (**A**) shows a Venn diagram of proteins identified in leghorn VEX, leghorn TEX and broiler TEX. Panel (**B**) shows a Venn diagram of datasets compared to the Vesiclepedia database of proteins [37], see also Appendix A). Panel (**C**) shows the Funrich enrichment of subcellular localization of proteins identified in leghorn VEX, TEX and broiler TEX.

**Table 1 genes-10-00116-t001:** Exosome Serum Sources, small RNA, microRNA (miRNA) and mRNA mapping data.

Serum Exosome Source	Bird Tag#	Raw Reads	Mapped to *Gallus gallus*	%	miRNAs Mapped to MDV	%	mRNAs Mapped to MDV	%
**leghorn VEX**	BL3815	27,168,090	5,539,665	20.4	478	0.009	11,995	0.22
	BL3841	16,544,388	12,264,873	74.1	224	0.002	29,586	0.24
	BL3850	11,567,549	73,94,774	63.9	6895	0.09	36,084	0.49
	BL3825	8,516,052	4,885,568	57.4	1864	0.04	8335	0.17
	**Means**	**15,949,020**	**7,563,369**	**47.4**	**2365**	**0.03**	**21,500**	**0.28**
**leghorn TEX**	OR1810	18,024,380	9,038,453	50.15	47,893	0.53	64,636	0.72
	OR1760	13,826,014	8,725,701	63.11	4126	0.05	314	0.003
	OR1838	26,009,675	8,559,904	32.91	4399	0.05	534	0.006
	OR748	16,422,100	8,321,176	50.67	9594	0.12	56,629	0.68
	**Means**	**18,570,542**	**8,661,309**	**46.64**	**16,503**	**0.19**	**30,528**	**0.35**
**broiler TEX**	BL4178	15,878,894	9,747,686	61.39	22,533	0.23	57,285	0.58
	BL4047	16,472,545	12,121,502	73.59	89,040	0.73	66,270	0.54
	BL4062	14,407,601	9,424,620	65.41	148,011	1.57	55,003	0.58
	BL4183	26,533,369	16,334,950	61.56	72,162	0.44	87,058	0.53
	**Means**	**18,323,102**	**11,907,190**	**65.0**	**82,937**	**0.70**	**66,404**	**0.55**
	**Totals**	**211,370,657**	**112,358,872**	**53.16**	**409,687**	**0.36**	**473,729**	**0.42**

VEX: vaccine-associated exosomes; TEX: tumor-associated exosomes; MDV: Marek’s disease virus.

**Table 2 genes-10-00116-t002:** *Gallus* miRNA read counts and miRNA abundance in leghorn VEX and TEX.

*G. gallus* miRNA	*p*-Value	Log_2_(VEX/TEX)	Leghorn TEX	Leghorn VEX
gga-miR-146b-5p	7.21 × 10^−5^	5.06	2046	68,424
PC-3p-53 200112	1.35 × 10^−3^	5.01	1134	36,566
gga-miR-10b-5p	2.03 × 10^−3^	4.38	21,594	449,498
gga-miR-2188-5p	3.24 × 10^−2^	3.65	16,457	207,103
gga-miR-27b-3p	3.24 × 10^−4^	3.63	8156	100,917
gga-miR-99a-5p	4.96 × 10^−2^	2.46	200,908	1,105,652
gga-miR-26a-5p	4.67 × 10^−3^	2.17	10,858	48,877
gga-miR-146c-5p	1.49 × 10^−3^	1.89	142,564	526,725
gga-miR-24-3p	7.26 × 10^−2^	1.43	3313	8949
gga-miR-146a-5p	3.91 × 10^−2^	1.16	18,000	40,291
gga-miR-23b-3p	6.65 × 10^−2^	1.16	3365	7535
gga-miR-30e-5p	4.99 × 10^−2^	0.88	4247	7789
gga-miR-199-3p	4.71 × 10^−2^	0.66	12,166	19,208
gga-miR-142-5p	3.59 × 10^−2^	−1.16	11,024	4947
gga-miR-125b-5p	4.55 × 10^−2^	−1.32	116,830	46,809
gga-miR-181a-5p	5.70 × 10^−2^	−1.44	31,255	11,509
gga-let-7	1.28 × 10^−2^	−1.83	20,148	5681
gga-miR-363-3p	2.27 × 10^−2^	−1.89	37,382	10,120
gga-miR-92-3p	4.55 × 10^−3^	−2.17	60,566	13,491
gga-miR-101-3p	3.40 × 10^−3^	−2.23	9426	2013
gga-let-7g-5p	6.86 × 10^−4^	−2.24	11,384	2417
gga-miR-451	2.47 × 10^−3^	−2.33	588,775	117,350
gga-miR-148a-3p	6.94 × 10^−3^	−2.44	113,250	20,858
gga-miR-21-5p	4.20 × 10^−3^	−2.88	120,353	16,302
gga-miR-7	1.86 × 10^−3^	−3.21	47,490	5122
gga-miR-15c-5p	1.50 × 10^−2^	−3.39	5628	537
gga-miR-16-5p	4.77 × 10^−4^	−3.45	10,178	934
gga-miR-16c-5p	7.74 × 10^−5^	−4.03	20,103	1231

**Table 3 genes-10-00116-t003:** *Gallus* miRNA read counts and miRNA expression in VEX and broiler TEX.

*G.**gallus* miRNA	*p-*Value	Log_2_(VEX/TEX)	Broiler TEX	Leghorn VEX
gga-miR-99a-5p	1.08 × 10^−2^	3.58	92,713	1,105,652
gga-miR-2188-5p	3.56 × 10^−2^	3.52	18,115	207,103
gga-miR-10b-5p	1.73 × 10^−2^	2.89	60,577	449,498
tgu-miR-1388-5p	1.20 × 10^−2^	1.81	14,403	50,359
gga-let-7f-5p	8.60 × 10^−3^	1.70	4454	14,519
gga-miR-146b-5p	1.58 × 10^−2^	1.70	21,014	68,424
gga-miR-26a-5p	2.81 × 10^−2^	1.53	16,930	48,877
gga-miR-27b-3p	1.99 × 10^−2^	1.49	35,868	100,917
gga-miR-30e-5p	3.23 × 10^−2^	1.41	13,857	36,839
gga-miR-146c-5p	8.41 × 10^−3^	1.34	208,239	526,725
gga-miR-23b-3p	6.49 × 10^−2^	1.06	3625	7535
gga-miR-30e-5p	8.75 × 10^−2^	0.82	4423	7789
gga-miR-24-3p	4.70 × 10^−2^	0.72	5445	8949
gga-miR-199-3p	8.75 × 10^−2^	−0.53	27,729	19,208
tgu-miR-425-5p	3.19 × 10^−2^	−0.78	5509	3198
gga-miR-92-3p	9.08 × 10^−2^	−0.85	24,402	13,491
gga-*let*-7i	2.78 × 10^−2^	−0.86	10,306	5,681
gga-miR-181a-5p	3.78 × 10^−2^	−0.88	21,235	11,509
gga-miR-126-3p	7.58 × 10^−2^	−1.05	10,493	5052
gga-miR-148a-3p	4.52 × 10^−2^	−1.15	46,178	20,858
tgu-miR-363-3p	6.53 × 10^−3^	−1.31	25,070	10,120
gga-miR-21-5p	4.23 × 10^−3^	−1.35	41,470	16,302
gga-miR-122-5p	2.98 × 10^−2^	−1.56	13,321	4527
gga-miR-142-5p	4.89 × 10^−3^	−2.08	20,957	4947
gga-miR-199-5p	1.00 × 10^−2^	−2.18	77,490	17,139
gga-miR-140-3p	1.07 × 10^−4^	−2.19	10,748	2359

**Table 4 genes-10-00116-t004:** Expression levels of MDV-encoded miRNAs in TEX compared to VEX.

		Leghorn TEX/Leghorn VEX		Broiler TEX/Leghorn VEX	
miRNA Origin Cluster	MDV-1-miR-	Fold Change	*p-*Value	Fold Change	*p-*Value
***meq* cluster 1**	**-M9**	9.41	0.22	39.43	0.07
	**-M5**	1.47	0.60	15.49	0.02
	**-M12**	3.44	0.37	49.00	0.08
	**-M3**	13.04	0.26	43.70	0.07
	**-M2**	2.19	0.39	40.60	0.03
	**-M4**	10.36	0.33	28.99	0.03
***meq* cluster 2**	**-M11**	4.27	0.16	69.20	0.01
	**-M31**	0.00	0.19	-	-
	**-M1**	1.20	0.89	19.10	0.04
**LAT cluster**	**-M8**	3.85	0.02	13.83	0.08
	**-M13**	4.33	0.32	10.78	0.05
	**-M6**	7.63	0.28	41.15	0.04
	**-M7**	22.45	0.16	37.38	0.01
	**-M10**	2.18	0.36	27.65	0.06

**Table 5 genes-10-00116-t005:** Serum VEX-overrepresented proteins.

Protein Name (Accession#)	VEX (NI) ^1^	TEX (NI) ^1^	RatioVEX/TEX	*p*-Value
Protocadherin-17 (XP_015132312.1)	2.31	0	11,109:1	0.015
Collagen -1 (XXII) chain (XP_015138660.1)	1.13	0	10,825:1	0.048
IgGFc-binding protein-like protein (XP_428412.5)	0.6	0	2851:1	0.035
Insulin-like growth factor-binding protein complex acid labile subunit isoform (XP_425222.3)	0.24	0	1159:1	0.014
Tenascin isoforms (XP_015134827.1)	0.15	0	998:1	0.014
Insulin-like growth factor-binding protein complex acid labile subunit isoform (XP_425222.3)	0.24	0	851:1	0.043
Muellerian-inhibiting factor precursor (NP_990361.1)	0.21	0	733:1	0.043
Collagen -1 (XXII) chain (XP_015138660.1)	1.13	0.04	27:1	0.039
Transferrin Receptor (XP_015146868.1)	0.1	0.01	12:1	0.005
C-reactive protein, pentraxin-related precursor (NP_001300649.1)	2.13	0.43	4.9:1	0.014

^1^—normalized intensity; Leghorn VEX/TEX (top 5 Leghorn VEX/Broiler TEX (bottom 5).

**Table 6 genes-10-00116-t006:** Serum TEX-overrepresented proteins.

Protein Name (Accession#)	VEX (NI) ^1^	TEX (NI) ^1^	Ratio TEX/VEX	*p*-Value
Histone H4 (XP_425458.2)	0	9.94	44,448:1	0.015
-2-antiplasmin isoform X2 (XP_015151528.1)	0	4.36	19,487:1	0.014
HSP90-α (NP_001034377.1)	0	3.12	13,971:1	0.029
**Pantetheinase isoform X1 (XP_015139593.1)**	0	2.77	12,388:1	0.005
Peroxiredoxin-1 (NP_001258861.1)	0	2.46	11,102:1	0.043
Deleted in malignant brain tumors 1 protein-like isoform X2 (DMBTL) (XP_015156078.1)	0	11.8	52,741:1	0.048
Ceruloplasmin (XP_015147339.1)	0	1.85	8265:1	0.014
**Pantetheinase isoform X3 (XP_015139595.1)**	0	1.51	6734:1	0.013
Clusterin precursor (NP_990231.1)	0	1.3	5815:1	0.013
Mannose-binding lectin precursor (NP_989680.2)	0	1.23	5515:1	0.01

^1^—normalized intensity; Leghorn TEX (top 5), Broiler TEX (bottom 5).

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
