# Peer review of "Comparison of the Transcriptomes and Proteomes of Serum Exosomes from Marek’s Disease Virus-Vaccinated and Protected and Lymphoma-Bearing Chickens"

_genes, 2019, doi:10.3390/genes10020116_

Round 1
Reviewer 1 Report
Wow this is a well written manuscript but a bit long. It was very interesting and will be welcoming by the herpesvirus community. Here are some suggestions:
Table ! legend mention mappable ot MDV and Gallus gallus genomes.
Figure 1 The sentence Panel B shows the nanoparticle size....should be changed to Panel C
shows the nanoparticle...
Page 9 line 381 put a space between G. and gallus
Page 9 line 383 needs a comma 485,724
Page 10 line 385 get ri of the period after of.
Page10 line 391 put a comma in 3,055
Page 16 line 477 What are gene symbols????
Page 25 line 637 put a comma after TEX and A period after respectively.
Page 28 line 735 put a space between G. and gallus
Page 31 line 872 get rid of reference 11. It is not needed.
Page 32 line 899 move parenthesis to after cells
Author Response
See attached document.

Reviewer 2 Report
General Comment:
In this manuscript, the authors investigated the exosomes in the serum of Marek’s disease virus infected and vaccinated chickens. They determined the miRNA content in exosomes of vaccinated (VEX) and tumor-bearing chickens (TEX). In addition, they assessed the mRNA and protein content of these exosomes. The authors provide an incredible amount of data that could have been split up into several manuscripts and provide information that is very important to the field. Due to the large amount of data, it is very hard to follow parts of the manuscript. The key messages of the manuscript are also buried in the huge data sets and could be further emphasized. This reviewer is convinced that this work is an important contribution to the field, however, several points should be addressed prior to publications.
Major Points:
1) It would be great if the authors could start each paragraph of the Results section with a sentence of two on the rational of the experiment/analysis. In addition, a sentence at the end of each paragraph summarizing the key findings would make the story more comprehensive.
2) The results section is in some parts very hard to follow as it lists many miRNAs that are up- or downregulated. These miRNAs are already included in the figure and are not needed in the text. The authors should rather focus on the take home message than listing miRNAs that do not mean anything to most readers.
3) One important finding is that exosomes from vaccinated animals have a plethora of viral mRNAs, while very are found in exosomes from tumor bearing animals. Unfortunately, the reader is flooded with so much information that this info does not get the attention it deserves. The authors should consider to focus on the key aspects and rather move less important information into the supplementary data.
4) The authors performed an interesting proteome analysis of VEX and TEX. Unfortunately, the key message e.g. the differences in the content of viral proteins (ICP4 vs. UL36/47) is lost as the text lists too many protein names/acronyms to follow. The authors should avoid including all the protein names in the text, as they are already in the tables, and rather focus on the key message. Focus on the biomarkers and leave out the less important information.
Minor Points:
1) The font size in several figures is too small to read on a printout (e.g. Fig. 1A-C, 2D, 3, 4B-E aso.). Please make sure that everything is readable in a printed version. In addition, some figures are too small to see (e.g. Fig. 6).
2) Table 1 should be moved into the supplementary data.
3) Figure 1C: is the size distribution shown in Figure 1C from a TEX or VEX sample. Optimally show the distribution for both.
4) Table 5 and 6 should be moved into the supplementary data.
5) Figure 6: the KEGG pathway analysis does not contribute much to the message of the manuscript. The figure itself is not readable and does not provide valuable information for most readers. If the authors insist to keep it then please move it to the supplementary data.
6) Figure 8 and 9 could/should be moved into the supplementary data.
7) Line 43f: The authors state that “An innate immune response mounted in response to early MDV lytic replication drives the establishment of latency”. This is not correct as adaptive responses mostly drive the establishment of latency.
8) Line 339: “incidence of MD was somewhat lower in unvaccinated contacts 339 compared to the inoculated shedders”. This is not needed as contact animals often have a lower MD incidence as they get infected later in life.
9) The discussion of 7 pages is a bit extensively and could be shortened. The authors try to put everything into a broader context, however, some of the comparisons e.g. to human cancer are very speculative, as there are large differences between humans and chickens and information is often not transferable.
